# General Perceptions and Knowledge of Antibiotic Resistance and Antibiotic Use Behavior: A Cross-Sectional Survey of US Adults

**DOI:** 10.3390/antibiotics12040672

**Published:** 2023-03-29

**Authors:** Caitlin M. McCracken, Kendall J. Tucker, Gregory B. Tallman, Haley K. Holmer, Brie N. Noble, Jessina C. McGregor

**Affiliations:** 1Department of Pharmacy Practice, Oregon State University College of Pharmacy, Portland, OR 97331, USA; 2Department of Pharmacy Practice, Nesbitt School of Pharmacy, Wilkes University, Wilkes-Barre, PA 18766, USA; 3School of Pharmacy, Pacific University, Hillsboro, OR 97123, USA; 4School of Public Health, Oregon Health & Science University-Portland State University, Portland, OR 97201, USA

**Keywords:** antibiotic resistance, public health, antibiotic use behavior

## Abstract

This study aimed to assess understanding of antibiotic resistance and evaluate antibiotic use themes among the general public. In March 2018, respondents that were ≥21 years old and residing in the United States were recruited from ResearchMatch.org and surveyed to collect data on respondent expectations, knowledge, and opinions regarding prescribing antibiotics and antibiotic resistance. Content analysis was used to code open-ended definitions of antibiotic resistance into central themes. Chi-square tests were used to assess differences between the definitions of antibiotic resistance and antibiotic use. Among the 657 respondents, nearly all (99%) had taken an antibiotic previously. When asked to define antibiotic resistance, the definitions provided were inductively coded into six central themes: 35% bacteria adaptation, 22% misuse/overuse, 22% resistant bacteria, 10% antibiotic ineffectiveness, 7% body immunity, and 3% provided an incorrect definition with no consistent theme. Themes that were identified in respondent definitions of resistance significantly differed between those who reported having shared an antibiotic versus those who had not (*p* = 0.03). Public health campaigns remain a central component in the fight to combat antibiotic resistance. Future campaigns should address the public’s understanding of antibiotic resistance and modifiable behaviors that may contribute to resistance.

## 1. Introduction

The increasing rate of antibiotic resistance is a global public health crisis that has been identified as one of the greatest threats to human health worldwide [1,2]. The emergence of antibiotic resistance has led to increasingly difficult-to-treat infections with limited or non-existent therapeutic options. The World Health Organization reports high levels of resistance globally to antibiotics that are used to treat bloodstream infections caused by *Acinetobacter* spp. (>56% resistant) and *Klebsiella pneumonia* (>57% resistant); 20% of *Escherichia coli*, the primary pathogen causing urinary tract infections, were resistant to first- and second-line urinary anti-infectives; also, gonorrhea, for which few treatment options remain, are now 60% resistant to ciprofloxacin. Antibiotic-resistant infections have been associated with increased morbidity and mortality; one study estimated that in 2019, there were 1.27 million deaths attributable to antibiotic-resistant bacterial infections [3]. In the United States alone, it is estimated that antibiotic-resistant bacteria cause more than 2.8 million infections and more than 35,000 deaths each year [2]. The cost burden that is associated with antibiotic-resistant infections in the US health system exceeds 2.9 billion dollars annually [4]. The increasing prevalence and economic burden signals the public health crisis that is posed by antibiotic resistance.

A study of 2010 to 2011 healthcare data identified that antibiotics were prescribed during 154 million ambulatory care visits in the United States, representing 12.6% of all ambulatory visits during the timeframe [5]. More than 60% of prescribed antibiotics were broad-spectrum in nature, and approximately 30% of all use was inappropriate. To slow the emergence of antibiotic resistance, judicious use of antibiotics is essential. In 2020, the White House released an updated National Action Plan for Combating Antibiotic-Resistant Bacteria with a five-year goal to slow the emergence of resistant bacteria and prevent the spread of resistant infections [6]. Strengthening educational programs to modify antibiotic use behavior and increase public awareness of antibiotic resistance was set as an objective to accomplish this goal. Prior to the 2015 National Action Plan, several public health initiatives, such as the Get Smart: Know When Antibiotics Work campaign and US Antibiotic Awareness Week, had been mobilized to focus on educating the public on appropriate use of antibiotics and the dangers that are associated with antibiotic-resistant infections.

Despite the focus placed on public health education campaigns, research linking these efforts to changes in antibiotic use behavior is limited. Misconceptions surrounding antibiotics contribute to inappropriate use behavior, such as sharing of antibiotics, which in turn drives increasing antibiotic resistance [7]. Historically, research to address inappropriate use has largely focused on inappropriate prescribing and not on inappropriate antibiotic use behavior [8]. The purpose of this survey is to assess knowledge and understanding of antibiotic resistance among the public and evaluate antibiotic use themes in association with understanding of antibiotic resistance.

## 2. Results

Of 1000 subjects that were invited to complete an inclusion screening form and participate in the survey panel, 770 persons responded to the invitation and 657 of these individuals met the inclusion criteria and completed the survey. Among the survey respondents, 45/50 US states were represented, with a majority of respondents living in a suburban geographic area. A total of 35% of respondents were aged 60 or older, 78% were female, and 89% identified as white (Table 1). Approximately 45% of the participants had a master’s degree or higher.

Nearly all respondents (99.5%) reported having taken an antibiotic in their lifetime, and over half of participants reported that they were moderately to extremely concerned about antibiotic resistance (Appendix A). A total of 7% of respondents believed antibiotics killed viruses.

Table 2 compares respondent characteristics and antibiotic use behaviors and antibiotic resistance themes. When asked to define antibiotic resistance, 10% characterized antibiotics as ineffective, 35% of participants described bacteria adapting, 22% focused on the mis-/overuse of antibiotics, 22% did not expand beyond “resistant bacteria”, 7% identified the person as having antibiotic immunity, and 3% provided an incorrect definition with no other consistent theme. A total of 18% of respondents reported having shared an antibiotic, which includes either receiving antibiotics or providing their own prescription antibiotic to another person. A total of 23% of respondents had obtained or knew someone else who had obtained antibiotics without a prescription and 47% had asked a doctor for an antibiotic. Only 44% of respondents reported that their doctor had ever talked with them about antibiotic resistance.

Respondents without a bachelor’s degree or higher were more likely to have shared an antibiotic (28%) compared to respondents with a bachelor’s degree or higher (15%, *p* < 0.01, Table 2). When asked to define antibiotic resistance, respondents with a bachelor’s degree or higher were more likely to describe bacteria mutating and less likely to describe non-informative or incorrect themes than respondents with below a bachelor’s degree level of education (*p* < 0.01). Respondents that were under 40 years of age, were more likely to have obtained or knew someone that had obtained an antibiotic without a prescription (33%) compared to respondents that were aged 40 and over (20%, (*p* < 0.01); yet, there was a larger proportion of respondents under 40 that accurately described resistance as bacteria adapting (46%) compared to participants 40 years or older (31%, *p* < 0.01).

Themes that were identified in respondent definitions of resistance differed significantly between those who reported having shared an antibiotic versus those who had not (*p* = 0.03); respondents who reported sharing an antibiotic were more likely to describe resistance with themes of antibiotic ineffectiveness (17% vs. 8%) and person immunity (10.5% vs. 7%) but were less likely to describe antibiotic misuse (17% vs. 23%), compared to those who did not report sharing antibiotics (Table 3). Respondents that had asked a doctor for antibiotics were more likely to describe misuse or overuse themes (25% vs. 19%) but were also more likely to report person or body immunity themes (11% vs. 4%) compared to those that had not asked a doctor for an antibiotic (*p* < 0.01). There were no significant differences in antibiotic resistance themes between those that had obtained or knew someone else that had obtained an antibiotic without a prescription.

There was no association between sharing or obtaining an antibiotic and talking to one’s doctor about antibiotic resistance (Table 4). Respondents that asked for an antibiotic either for themselves or their child were more likely to have discussed antibiotic resistance with their doctor than those who had not asked for antibiotics (58% vs. 39%, *p* < 0.01).

## 3. Discussion

Global reports from the United Nations council and the World Health Organization continue to stress that raising awareness through public health campaigns is critical towards combating antibiotic resistance [9,10,11]. Currently, there remains a limited understanding of the concept of antibiotic resistance, and the best methods for providing education that does not dissuade necessary use are unknown. In our survey, we sought to examine associations between respondent characteristics, antibiotic use behavior, and understanding of antibiotic resistance to inform future targets for public health campaigns.

Prior public health campaigns to educate the public regarding the effectiveness of antibiotics against bacterial versus non-bacterial infections were successful to a certain extent [9]. Nearly all respondents in the sample had been prescribed an antibiotic and, overall, were more educated than the general public [12]. Within our study sample, only 7% believed antibiotics killed viruses. This may have been due to respondents having attained higher levels of education than the general US population. A cross-sectional survey in community pharmacies in Kansas in 2018 that was conducted by Seipel et al. found that those with higher education were less likely to believe antibiotics killed viruses (20.9% vs. 43.1%) [13]. Approximately 45% of participants in our population had a master’s degree or higher, which is considerably higher than the US national average of 12–13% [12]. Nevertheless, among respondents with a bachelor’s degree or higher, approximately one-third either incorrectly defined antibiotic resistance or provided a neutral/non-informative definition without elaboration. A total of 22% of participants described the human contribution of misuse or overuse of antibiotics in the spread of resistance, but 18% of all respondents engaged in behaviors that may increase the risk of antibiotic resistance by sharing their antibiotic prescription. Public health campaigns should be recognized for their importance in educating the public; however, disconnects may still exist between knowledge and patient behaviors. Thus, campaigns place a greater focus on educating the public on the harms that are associated with antibiotic resistance and their role in preventing antibiotic misuse, which exacerbates the problem of increasing antibiotic resistance. Campaigns have focused heavily on ‘complete the course’ but a shift towards’ do not start the course’ if not prescribed to you may better address the problem, particularly in countries where antibiotics can be accessed without a prescription [14].

Among respondents that requested an antibiotic, a little more than half had a doctor counsel them on antibiotic resistance. This suggests that prescribers are more likely to have conversations with patients about antibiotic resistance if the patient is seeking antibiotics. However, there was no significant difference between those that had a conversation about antibiotic resistance with their doctor and antibiotic resistance definition themes. Despite this, the patient-doctor conversation should be viewed as an important educational opportunity for discussing antibiotic resistance. Doctor patient conversations should focus on a clear definition of resistance and individual responsibilities of appropriate antibiotic use, regardless of whether or not antibiotic seeking behaviors are identified during the visit. Small interactions such as this play a significant role in increasing awareness of antibiotic resistance and should be a target for public health campaigns.

The survey collected retrospective, self-reported data, which is subject to reporting and recall bias and, as such, may underestimate the true incidence of antibiotic use behaviors that were reported in this study. Participants self-selected to participate in potential studies with ResearchMatch.org and also self-selected to take part in our study. Participants were limited to English speaking US adults with access to a computer and the internet. Respondents included a higher proportion of age >40, females, white race, and received more education than the general population, thus these data may not be generalizable to other demographic groups. Further research is needed to better understand the antibiotic resistance knowledge and antibiotic use behaviors in younger populations, non-white racial groups, men, and persons who have not received a bachelor’s degree or higher levels of education. Still, this work highlights the ongoing need for continued effective public health education campaigns around antibiotic use.

The SARS-CoV-2 pandemic has shown that previous methods for public health education may have been ineffective at preparing the general public for managing COVID-19 [15]. The utilization of social media as information sources may contribute to the spread of misinformation regarding infectious diseases and underscores the increased need for effective public health education. During the pandemic, public health agencies have strived to educate the public regarding general knowledge of viruses, disease transmission, prevention, and treatment. Although our survey was conducted prior to the outbreak of COVID-19, the concepts and information provided are important for informing future public health education campaigns to increase effectiveness of public health education.

## 4. Materials and Methods

### 4.1. Study Design and Participants

We conducted a cross-sectional survey study of participants from across the United States. The participants were recruited from ResearchMatch.org, a national online clinical registry that matches participants that are interested in participating in research with researchers at institutions that have received research support from the US National Institute of Health’s Clinical and Translational Science Awards program. Inclusion was limited to English speaking participants 21 years or older that resided in the US with access to the internet. An email was sent to a random sample of 1000 eligible participants, via ResearchMatch.org inviting them to participate in a survey panel on combating antibiotic-resistant bacteria (CARB) during March 2018. If the respondents agreed to participate in the panel, they were sent an inclusion/exclusion criteria screening form, a study information sheet, and notification that completion of the survey indicated consent to participate in this study.

### 4.2. Survey Design

The survey questions that were used in this study were developed by the study team to understand participant’s expectations, knowledge, and opinion regarding antibiotic prescribing and understanding of antibiotic resistance (see Appendix A). The survey questions underwent internal usability testing and revision prior to finalizing; the questions were organized by subject matter content. Surveys were provided through our institution’s instance of REDCap, a secure clinical research compliant web-based platform for collecting, managing, and storing research study data [16,17]. As an incentive, the first 100 survey respondents were each given a $5 gift card. This research was approved by the Oregon Health and Science University institutional review board.

### 4.3. Data Analysis

The participants were asked to define antibiotic resistance as an open-ended survey question. Inductive content analysis was used to create central themes from the data. There were two rounds of thematic coding that were performed by a qualitative analyst; the first round reduced responses into phrases that were used by the respondent to describe resistance and a second round of coding grouped phrases into broader shared themes. Developing and final thematic categories were reviewed by the study team to ensure they were mutually exclusive and an accurate representation of the respondent’s definition. There were five mutually exclusive themes that were identified from the participant responses. Three categories reflect accurate concepts that were attributed to antibiotic resistance: ineffective antibiotics, bacteria adapting, and mis-/overuse of antibiotics. One category captures neutral responses with no other theme than “resistant bacteria” and two categories reflect incorrect responses, the person/body having antibiotic immunity and incorrect with no other consistent theme among respondents to be categorized beyond incorrect (e.g., virus becomes immune).

Descriptive statistics (proportions, medians, etc.) were used to describe participant characteristics. Chi-square tests were used to compare proportional differences in antibiotic use behaviors by respondent characteristics as well as antibiotic resistance definitions. *p* < 0.05 was considered statistically significant. Statistical analysis was performed using SAS v. 9.4 (SAS Institute, Inc.; Cary, NC, USA).

## 5. Conclusions

Awareness of antibiotic resistance is growing at a rate much slower than the threat and consequences of antibiotic misuse. We must modify person-level antibiotic use behaviors that contribute to the spread of resistance. Public health campaigns remain a central component in the fight to combat antibiotic resistance, however the approach remains difficult, particularly given the complexity of the topic. Targeted campaigns for younger demographics and those without post-secondary education can go a long way in the fight to combat antibiotic resistance. These targeted campaigns should place greater focus on patient-provider conversations regarding the appropriate use of antibiotics and resistance. Finally, future research and public health initiatives should address patient motives for antibiotic requests and sharing. Given the global burden of infectious diseases and the morbidity and mortality associated with antibiotic-resistant infections, it is critical to engage the public as well as healthcare providers and public health officials in the global battle against antibiotic resistance.

## Figures and Tables

**Table 1 antibiotics-12-00672-t001:** Respondent characteristics (*n* = 657).

	*n*	%
Age (yrs)		
21–39	183	28%
40–59	242	37%
60+	232	35%
Female	515	78%
Race		
White	587	89%
Black	30	5%
Asian	13	2%
Other ^a^	18	4%
Hispanic	18	3%
Geographic Area		
Urban	182	28%
Suburban	364	56%
Rural	101	15%
Other	6	1%
Highest Level of Education		
≤High School Graduate	19	3%
Some College/Associate’s Degree	124	19%
Bachelor’s Degree	215	33%
Master’s Degree	181	27%
Doctorate/Professional Degree	117	18%
Has Children ^b^	383	59%
Healthcare decision maker for older adult	212	32%

^a^ Other incudes I Native Hawaiian or Pacific Islander, 3 American Indian/Alaskan Native, 14 Other; ^b^ Includes 2 expectant respondents.

**Table 2 antibiotics-12-00672-t002:** Respondent characteristics and antibiotic use behaviors.

	Overall *n*/657 (%)	<Bachelor’s Degree *n*/143 (%)	≥Bachelor’s Degree *n*/510 (%)	*p* Value ^+^	Age < 40 *n*/183 (%)	Age ≥ 40 *n*/473 (%)	*p* Value ^^^
Has shared an antibiotic (either received it or provided it).	119/655 (18%)	40/143 (28%)	79/510 (15%)	*p* < 0.01	36/183 (20%)	83/471 (18%)	*p* = 0.54
Has obtained or knows someone who has obtained antibiotics without a prescription?	154/656 (23%)	39/143 (27%)	115/511 (22.5)	*p* = 0.23	60/183 (33%)	94/472 (20%)	*p* < 0.01
Has asked a doctor for an antibiotic (self or for child)?	309/656 (47%)	78/143 (55%)	231/511 (45%)	*p* = 0.05	79/183 (43%)	230/472 (49%)	*p* = 0.20
Antibiotic Resistance Themes ^a^							
Ineffective antibiotics	61/635 (10%)	12/129 (9%)	49/504 (10%)	*p* < 0.01	11/177 (6%)	50/458 (11%)	*p* < 0.01
Bacteria adapts/mutates	223/635 (35%)	33/129 (26%)	189/504 (37%)		82/177 (46%)	141/458 (31%)	
Misuse or overuse of antibiotics	142/635 (22%)	24/129 (19%)	117/504 (23%)		33/177 (19%)	109/458 (24%)	
Person/body immunity to antibiotics	46/635 (7%)	13/129 (10%)	33/504 (7%)		5/177 (3%)	41/458 (9%)	
Resistant bacteria no elaboration, neutral	141/635 (22%)	36/129 (28%)	105/504 (21%)		38/177 (21%)	103/458 (22%)	
Incorrect w/no consistent theme	22/635 (3%)	11/129 (8%)	11/504 (2%)		8/177 (5%)	14/458 (3%)	

^+^ Respondents without a bachelor’s degree compared to respondents with a bachelor’s degree or higher; ^^^ respondents under the age of 40 compared to respondents aged 40 and over; ^a^ 22 respondent answers were not included because they either skipped the question or did not define resistance.

**Table 3 antibiotics-12-00672-t003:** Association between antibiotic resistance themes and antibiotic use behaviors.

Antibiotic Resistance Themes ^a^	Shared (Received/Provided) *n* (%) ^b^				Obtained (Self or Other) *n* (%) ^c^				Asked Doctor for Antibiotics *n* (%) ^d^			
	Yes (*n* = 113)		No (*n* = 520)		Yes (*n* = 151)		No (*n* = 483)		Yes (*n* = 301)		No (*n* = 333)	
Ineffective antibiotics	19	(17%)	42	(8%)	17	(11%)	44	(9%)	32	(11%)	29	(9%)
Bacteria adapts/mutates	37	(33%)	185	(35%)	61	(40%)	161	(33%)	90	(30%)	133	(40%)
Misuse or overuse of antibiotics	19	(17%)	122	(23%)	32	(21%)	110	(23%)	76	(25%)	65	(19%)
Resistant bacteria no elaboration	23	(20%)	118	(23%)	31	(21%)	110	(23%)	61	(20%)	80	(24%)
Person/Body immunity to antibiotics	12	(10.5%)	34	(7%)	6	(4%)	40	(8%)	32	(11%)	14	(4%)
Incorrect w/no consistent theme	3	(2.5)	19	(4%)	4	(3%)	18	(4%)	10	(3%)	12	(4%)

^a^ 22 respondent answers were not included because they either skipped the question or did not define resistance; Statistical significance determined by chi square test; ^b^
*p* = 0.03; ^c^
*p* = 0.32; ^d^
*p* ≤ 0.01.

**Table 4 antibiotics-12-00672-t004:** Association between doctor and respondent conversations about antibiotic resistance and antibiotic use behavior.

	Doctor Discussed Antibiotic Resistance		
	Yes (*n* = 290)	No/Not Sure (*n* = 367)	
Shared an antibiotic (either received it or provided it)	56 (19)	63 (17)	*p* = 0.48
Obtained or knows someone who has obtained antibiotics without a prescription	66 (23)	88 (24)	*p* = 0.70
Asked a doctor for an antibiotic (self or for child)	167 (58)	142 (39)	*p* < 0.01
Antibiotic Resistance Themes			*p* = 0.95
Ineffective antibiotics	27 (10)	34 (10)	
Bacteria adapts/mutates	94 (33)	129 (37)	
Miss or over use of antibiotics	66 (23)	76 (21)	
Person/body immunity to antibiotics	22 (8)	24 (7)	
Resistant bacteria no elaboration, neutral	65 (23)	76 (21)	
Incorrect, no consistent theme	9 (3)	13 (4)	

Statistical significance determined by chi square tests.

## Data Availability

The data presented in this study are available on request. The data are not publicly available due to patient privacy.

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
