# Peer review of "General Perceptions and Knowledge of Antibiotic Resistance and Antibiotic Use Behavior: A Cross-Sectional Survey of US Adults"

_antibiotics, 2023, doi:10.3390/antibiotics12040672_

Round 1

Reviewer 1 Report

1- make sure to use the font specified in the author instructions. e.g., email for corresponding author is larger font compared to the rest of the manuscript 

2-indicate study type (cross sectional study in the title)

3- in text citation should be based on author instruction, please use MDPI Antibiotic template with the reference manual 

4- the first section in the method paragraph should be about setting and design 

5- I don't know where is the discussion section start? there is no title names discussion 

6- use this article to structure your discussion, re-write the discussion 

Author Response

1- make sure to use the font specified in the author instructions. e.g., email for corresponding author is larger font compared to the rest of the manuscript

Response: Some of the questions raised pertain to the typeset that was applied when the manuscript was reformatted for reviewers by Journal staff.

2-indicate study type (cross sectional study in the title)

Response:  Will now identify the study type as a cross-sectional survey in the Methods. We have also modified the title.  The new title is:  General Perceptions and Knowledge of Antibiotic Resistance and Antibiotic Use Behavior - A Cross-sectional Survey of US Adults

3- in text citation should be based on author instruction, please use MDPI Antibiotic template with the reference manual

Response: We have updated the in-text citations and references using the antibiotic manuscript template

4- the first section in the method paragraph should be about setting and design

Response:  Thank you for the feedback, we have modified this section of the Methods to more clearly indicate that this was a cross-sectional survey study of participants across the United States, and we have provided additional details.

5- I don't know where is the discussion section start? there is no title names discussion

Response: Unfortunately, this was the result of a typeset issue that occurred when our document was formatted for reviewers by Journal editorial staff.

6- use this article to structure your discussion, re-write the discussion

Response: It appears the reviewer was referencing a specific article, however no article was attached to the reviewer comments we received.

Reviewer 2 Report

Many thanks for the opportunity to review this paper on the general perception on AMr and antibiotic use behaviour. I would like to congratulate the Authors on their research as this is an important topic to evaluate in the general population. The manuscript is well written. I recommend some revisions.

Title: should be more descriptive of the study, eg study design and location should be stated

Abstract: lines13-15 inclusion criteria of participants should also mention that participation was limited to the US.

Introduction:

I recommend this paper with an updated and global perspective on the burden of AMR: doi: 10.1016/S0140-6736(21)02724-0. 

lines 52-59 and discussion: some further context on public health campaigns in the US could be of interest for the international readership of the journal, particularly considering evaluating the impact of these campaigns is part of the aims of the study. 

Methods: this section should be expanded. How was the survey designed? In particular, how was the survey structured? It appears to have some resemblance to a KAP survey but knowledge, opinion and experience are grouped together. 

Anorher important point: How was the sample selected? lines 61: some further explanation on ResearchMatch.org could be useful, in particular concerning representativeness of the sample. How are participants identified and recruited? Is it possible to know what was the response rate among eligible participants? Are there any potential biases linked to using this platform? this should also be reflected in the limitations section.

Analyses: What definition was considered correct for AMR? How were themes identified?

lines 75-76 should be further explained, what behaviours were considered indicators of inappropriate use? 

Results: results concerning respiratory infections are lacking, even though this was an entire section of the survey.

Discussion

The paragraph on the impact of the pandemic could benefit of the addition of some discussion on the potential impact on AMR: https://doi.org/10.1093/jac/dkaa194, doi: 10.3390/antibiotics11050695. doi: 10.2807/1560-7917.ES.2020.25.45.2001886.

Author Response

Title: should be more descriptive of the study, eg study design and location should be stated

Response: We have edited the title of the manuscript to include this information.

Abstract: lines13-15 inclusion criteria of participants should also mention that participation was limited to the US.

Response: We have added that participants are limited to the United States.

Introduction:

I recommend this paper with an updated and global perspective on the burden of AMR: doi: 10.1016/S0140-6736(21)02724-0.

Response: Thank you for the feedback.  We have expanded our introduction section to include a more global perspective on antimicrobial resistance.

lines 52-59 and discussion: some further context on public health campaigns in the US could be of interest for the international readership of the journal, particularly considering evaluating the impact of these campaigns is part of the aims of the study.

Response: We have expanded the Introduction and Discussion sections to better describe the global relevance of this research.

Methods: this section should be expanded. How was the survey designed? In particular, how was the survey structured? It appears to have some resemblance to a KAP survey but knowledge, opinion and experience are grouped together.

Response: We have added additional information about the survey design to the Methods. The survey is also included in the appendix for better understanding of the specific questions asked and response format. The survey was not specifically designed to follow the KAP format/methodology.

Another important point: How was the sample selected? lines 61: some further explanation on ResearchMatch.org could be useful, in particular concerning representativeness of the sample. How are participants identified and recruited? Is it possible to know what was the response rate among eligible participants? Are there any potential biases linked to using this platform? this should also be reflected in the limitations section.

Response: We have added information to describe Researchmatch.org and how participants were selected to the Methods section. We have added information to the limitation section in the Discussion about the potential for selection bias that may arise from using this platform as a recruitment tool. This expanded Discussion includes text describing that participants were more educated than the general US public and yet understanding of antibiotic resistance was still suboptimal.

Analyses: What definition was considered correct for AMR? How were themes identified?

Response: We have added more information in the methods to describe themes identified as correct and incorrect. Please see the Data Analysis subsection in the Methods section.

lines 75-76 should be further explained, what behaviors were considered indicators of inappropriate use?

Response: Lines 75-76 refer to the statement, “Chi square tests were used to test proportional difference of antibiotic use behaviors and respondent characteristics as well as antibiotic resistance definitions”. We do not describe specific antibiotic use behaviors as indicators of inappropriate use, we only describe them as behaviors. Some behaviors could be considered inappropriate and some could be considered opportunities for education, but the individual behaviors are not described in terms of inappropriate or appropriate behaviors.

Results: results concerning respiratory infections are lacking, even though this was an entire section of the survey.

Response:   The respiratory infections survey was a separate survey that respondents were invited to participate in. We have removed the section referencing the respiratory infections survey at the suggestion of Reviewer 3. This removes any confusion.

Discussion

The paragraph on the impact of the pandemic could benefit from the addition of some discussion on the potential impact on AMR: https://doi.org/10.1093/jac/dkaa194, doi: 10.3390/antibiotics11050695. doi: 10.2807/1560-7917.ES.2020.25.45.2001886.

            Response.  We agree with the reviewer regarding the important impact of antimicrobial resistance. We have expanded our introduction to better describe the global burden of resistance. However, the reference provided by the reviewer focuses on antibiotic overuse during the COVID-19 pandemic, and focuses on antimicrobial stewardship issues within the context of hospitalized patients.  Much of our own research team’s research focuses on antimicrobial stewardship in hospital settings, and we also have a pending publication in another journal focused on this issue in the context of the pandemic.  Yet, the scope of this manuscript is considerably broader than focus of this survey and hence not an appropriate fit for the Discussion of our manuscript.

Reviewer 3 Report

First of all, I would like to thank you for giving me the opportunity to review this manuscript, which highlights what still needs to be done in terms of population health education on antibiotic use/abuse/misuse. The uploaded paper contains annotations of certain aspects to be completed, corrected or improved. I would also like to make some recommendations, with the
sole objective of improving the manuscript.
It is highly advisable that the authors follow the instructions for authors and order correctly the manuscript: Results section is before Material and methods, and this latter section is after Discussion.
References should also be completed/corrected, and format adapted to the journal guidelines.
Methods section
1. It is not indicated whether the participants signed any type of informed consent before answering the survey.
2. It should be better explained that the study is part of a larger study, and that only the part related to antibiotic resistance is shown here. Is it necessary to include in the appendix A those questions that are not referred to in the study?
3. It should be explained what ResearchMatch.org is for all non-U.S. citizens.
4. Is not it an ethical problem to pay to fill out a survey?
5. How were the questions selected? Were they taken from any previous surveys? Supporting references should also be included. Has any validation been done? All these points are critical, and the results of the study depend on them.
6. It is not explained how and by whom the coding of the responses was carried out. Was any specific methodology used?
7. How were data preserved, and was their anonymity guaranteed?
Limitations: not speaking English or not having access to the internet, is a limitation?

I have uploaded the manuscript pdf with some comments to clarify and correct.

Author Response

Thank you for your detailed feedback.  Please see below:

It is highly advisable that the authors follow the instructions for authors and order correctly the manuscript: Results section is before Material and methods, and this latter section is after Discussion.

Response: The manuscript we submitted was typeset by the Journal staff prior to submission to reviewers. We will work with the Journal to ensure that the manuscript is appropriately formatted and typeset. We have updated the manuscript using the journal's preferred template.

References should also be completed/corrected, and format adapted to the journal guidelines.

            Response: We have updated the References using the MDPI citation style.

Methods section

  1. It is not indicated whether the participants signed any type of informed consent before answering the survey.

Response: We have added a statement in the methods section to detail the use of an information sheet within the web-based survey that served as informed consent if they proceed to take the survey. No signed consents were collected. This was approved by our institution’s Institutional Review Board (IRB) prior to any study materials being sent to participants.

  1. It should be better explained that the study is part of a larger study, and that only the part related to antibiotic resistance is shown here. Is it necessary to include in the appendix A those questions that are not referred to in the study?

Response: We have removed the respiratory virus survey from the manuscript to reduce confusion. As stated above, subjects were asked to respond to two surveys, but data from the respiratory virus survey is not presented in this manuscript.

  1. It should be explained what ResearchMatch.org is for all non-U.S. citizens.

Response: We have added further information to describe ResearchMatch.org to the Methods section.

  1. Is not it an ethical problem to pay to fill out a survey?

Response: Providing an incentive to respond was reviewed by our IRB board (i.e., the ethics oversight group for human subjects research) and not deemed unethical. The practice of providing an incentive is widely used in survey research. Only the first 100 respondents received a gift card so not all participants were paid. In addition, the amount was limited to $5 dollars, which was not considered to be coercive or unethical by our IRB. 

  1. How were the questions selected? Were they taken from any previous surveys? Supporting references should also be included. Has any validation been done? All these points are critical, and the results of the study depend on them.

Response: We have expanded our description of the survey development in the Methods. Survey questions were developed by the study team. Most questions were developed to be broad, easily understandable to a wide audience and for specific research questions. Questions were not taken from previous surveys.  Antibiotic behaviors were developed with study pharmacists based on their experience working with patients. Some questions were asked for follow up in future surveys. Early drafts of the survey underwent usability testing with a convenience sample of members of the general public and used to refine the survey.

  1. It is not explained how and by whom the coding of the responses was carried out. Was any specific methodology used?

Response: We have added an additional paragraph in the methods section to describing how coding of antibiotic resistance definitions was performed.

  1. How were data preserved, and was their anonymity guaranteed?

Response: Data is preserved within REDCap which acts as a survey platform/data collection tool and repository. Only study team members have access to REDCap and analysis is performed by study team members using de-identified data. Data connecting panel participants to the survey data is stored separately. Anonymity is never guaranteed but per our information sheet provided to participants this is explained and how we take measures to prevent this. Data storage and security is reviewed by our IRB prior to any data collection taking place.

Limitations: not speaking English or not having access to the internet, is a limitation?

Response: This is a limitation in the generalizability of the survey as it does not reflect the experience or understanding of non English speaking respondents or those without internet. This is detailed in the limitation section.

Reviewer 4 Report

The manuscript assesses understanding of antibiotic resistance and evaluates antibiotic use among the general public. Considering the threat of antibiotic resistance, the data collected in this survey are important in the fight against the overuse or misuse of antibiotics. In that respect, this manuscript is interesting, suggesting that future campaigns should address public understanding of antibiotic resistance and the modifiable behaviors that can contribute to resistance. However, an important limitation of this survey is represented by the educational qualification of the sample, which is higher than that of the general population. Overall, the study is well written and current.

Author Response

Thank you for your kind feedback. 

Round 2

Reviewer 1 Report

No comments at this time

Author Response

Thank you for your feedback!

Reviewer 3 Report

1. Lines 78-80: It should be stated somewhere that the paragraph refers to questions listed in appendix A, as they do not appear in any table.

2. Although the authors have included information about the coding of responding in the Materials and Methods section, information is only partial. Please, describe in detail who carried out coding and how, or how consensus was reached if different themes were selected for an answer.

3. The authors have adapted the manuscript to the journal's format, but the protocol code and approval date are still missing (required according the journal's instructions).

Author Response

  1.     Lines 78-80: It should be stated somewhere that the paragraph refers to questions listed in appendix A, as they do not appear in any table.

 Response:  Thank you for this feedback, we have modified line 78 to 80 to refer to appendix A, “ Nearly all respondents (99.5%) reported having taken an antibiotic in their lifetime, and over half of participants reported they were moderately to extremely concerned about antibiotic resistance (Appendix A). Seven percent of respondents believed antibiotics killed viruses.”

  1. Although the authors have included information about the coding of responding in the Materials and Methods section, information is only partial. Please, describe in detail who carried out coding and how, or how consensus was reached if different themes were selected for an answer.

Response: Thank you for the feedback, we have added the following statement to lines 213-218. “Two rounds of thematic coding was performed by a qualitative analyst; the first round reduced responses into phrases used by the respondent to describe resistance and a second round of coding grouped phrases into broader shared themes. Developing and final thematic categories were reviewed by the study team to ensure they were  mutually exclusive and an accurate representation of the respondent’s definition.”

  1. The authors have adapted the manuscript to the journal's format, but the protocol code and approval date are still missing (required according the journal's instructions).

Response : We have added the IRB approval number and the approval date to the manuscript. OHSU study approval number 17118 on 1/23/2018.  Please see the updated Institutional Review Board Statement on lines 244-246